# ENHANCING NEURAL NETWORK INTERPRETABILITY WITH FEATURE-ALIGNED SPARSE AUTOENCODERS

## ABSTRACT

Sparse Autoencoders (SAEs) have shown promise in improving the interpretability of neural network activations, but can learn features that are not features of the input, limiting their effectiveness. We propose MUTUAL FEATURE REGULARIZATION (**MFR**), a regularization technique for improving feature learning by encouraging SAEs trained in parallel to learn similar features. We motivate MFR by showing that features learned by multiple SAEs are more likely to correlate with features of the input. By training on synthetic data with known features of the input, we show that MFR can help SAEs learn those features, as we can directly compare the features learned by the SAE with the input features for the synthetic data. We then scale MFR to SAEs that are trained to denoise electroencephalography (EEG) data and SAEs that are trained to reconstruct GPT-2 Small activations. We show that MFR can improve the reconstruction loss of SAEs by up to 21.21% on GPT-2 Small, and 6.67% on EEG data. Our results suggest that the similarity between features learned by different SAEs can be leveraged to improve SAE training, thereby enhancing performance and the usefulness of SAEs for model interpretability.

## 1 INTRODUCTION

Interpretability aims to explain the relationship between neural network internals and neural network outputs. Many interpretability techniques examine raw activations, equating proposed fundamental units of neural networks such as neurons or polytopes to human understandable concepts (Erhan et al., 2009; Nguyen et al., 2016; Bau et al., 2017; Olah et al., 2018; Black et al., 2022). These techniques often benefit from a clean correspondence between those fundamental units and concepts, and may fail if concepts are distributed over many units, or many concepts focused in a single unit, such as in the case of feature superposition (Elhage et al., 2022). We describe *features of the input* as the atomic, human-understandable concepts represented by input data.

To derive a representation of activations with a stronger one-to-one correspondence of features and neurons, sparse autoencoders (SAEs) have been trained on neural network activations. The decoders of SAEs trained on neural network activations have been shown to form dictionaries of features more easily explained than the neurons themselves, making SAEs potentially useful for understanding the internals of neural networks (Bricken et al., 2023; Cunningham et al., 2024; Gao et al., 2024).

Despite the recent popularity of SAEs, early results suggest they may learn features that are not features of the input, reducing their usefulness for interpretability (Till, 2024; Huben, 2024; Anders et al., 2024). One failure mode considers transformations on the space of inputs: features of the input that are 'split' over multiple decoder weights, or multiple features 'composed' in one decoder weight. Conceivably, the representation learned by the SAE could be so varied from the input as to contain features entirely incompatible with the input space. These failures are alarming, as studying an SAE would not be guaranteed to reveal information about the neural network that SAE was trained on activations from. In the worst case, if features were commonly split and composed, it is not obvious why studying SAEs would be more useful than studying the raw activations directly, although prior work has given evidence against this.

We hypothesize that if a feature is learned by multiple SAEs, that feature is more likely to be a feature of the input, and show that this is true for SAEs trained on synthetic data comprised of known features. Based on this result, we encourage multiple SAEs trained on the same data to learn common

Figure 1: Our experimental pipeline for training SAEs with MFR. In step one, we extract activations from a neural network, represented by the interconnected nodes on the left. These activations are the inputs for our SAEs. In step two, we train multiple SAEs on the extracted activations. Each SAE learns to reconstruct the input activations through a sparsity constraint on the hidden layer. MFR involves several steps: We first check for inactive features in the SAE hidden state after applying the TopK activation function. If too many inactive features are detected, we reinitialize the weights of the affected SAE. We also include an auxiliary penalty to encourage the SAEs to learn similar features, shown by the final text box.

features through conditionally reinitializing SAE weights, and an auxiliary penalty calculated using the similarity of the SAE weights. We name this reinitialization technique and auxiliary penalty MUTUAL FEATURE REGULARIZATION **(MFR)**.

Using SAEs trained with MFR, we learn more features of the input than baseline SAEs when training on synthetic data (Section 3). We then train SAEs with MFR on activations from GPT-2 Small (Radford et al., 2019) and on electronencephalography (EEG) data, showing that MFR improves SAEs at scale, and on real-world data (Section 4). Our findings indicate that MFR helps avoid features not in the input space and improves performance on key SAE evaluations, potentially increasing their usefulness for interpretability.

## 2 BACKGROUND

### 2.1 SPARSE AUTOENCODERS

Olshausen and Field (1996) introduced unsupervised learning of sparse representations, capturing structure in data more efficiently than dense representations. Sparse autoencoders (SAEs) have since found wide application in domains such as representation learning (Coates et al., 2011; Henaff et al., 2011), denoising (Vincent et al., 2010; Duan et al., 2014), and anomaly detection (Sakurada and Yairi, 2014; Xu et al., 2015).

SAEs reconstruct an input $\mathbf{x} \in \mathbb{R}^d$ through a hidden representation $\mathbf{h} \in \mathbb{R}^h$, minimizing the reconstruction loss $\|\mathbf{x} - \hat{\mathbf{x}}\|_2^2$ while maintaining sparsity in $\mathbf{h}$, written as $\hat{\mathbf{x}} = \mathbf{W}'\sigma(\mathbf{W}\mathbf{x} + \mathbf{b})$, where $\mathbf{W} \in \mathbb{R}^{h \times d}$ is the encoder weight matrix, $\mathbf{W}' \in \mathbb{R}^{d \times h}$ is the decoder weight matrix, $\mathbf{b} \in \mathbb{R}^h$ is the encoder bias, and $\sigma$ is an activation function on $\mathbf{h}$. Ideally columns of $\mathbf{W}'$ correspond to features comprising $\mathbf{x}$.

Recent work has shown that the TopK activation function (Makhzani and Frey, 2013) better approximates the L0 norm in training than alternative techniques such as L1 regularization (Gao et al., 2024), allowing more precise control over the sparsity of $\mathbf{h}$. The TopK activation function on $\mathbf{h}$ is defined as:

$$\sigma_k(\mathbf{h})_i = \begin{cases} h_i & \text{if } h_i \geq \tau_k(\mathbf{h}) \\ 0 & \text{otherwise} \end{cases}$$

where $\tau_k(\mathbf{h})$ is the $k$th largest activation in $\mathbf{h}$. We focus exclusively on SAEs with a TopK activation function, and use the transpose of the encoder weights as the decoder, halving the number of trainable parameters with minimal performance impact as shown by Cunningham et al. (2024).

## 2.2 EVALUATING SPARSE AUTOENCODERS

Evaluating SAEs is challenging due to the lack of a ground truth for the input features represented by large neural networks. Thus, SAE evaluations act as proxies the extent to which interpretable representation of these input features have been learned in a way that does not require access to them.

The reconstruction loss, measured as the Euclidean distance between the SAE input and output, is a widely used metric for the faithfulness of an SAE's learned representations to the input features represented by a neural network. Although reconstruction loss does not account for the interpretability of the learned features, improved reconstruction loss has previously been accompanied by improved performance on interpretability evaluations (Rajamanoharan et al., 2024a; Gao et al., 2024; Rajamanoharan et al., 2024b). However, the reconstruction loss is not itself sufficient to evaluate SAEs, as there may be solutions to optimizing reconstruction loss that do not preserve the structure of the input features or convey information about them, such as learning the identity function.

The interpretability of SAE features has been evaluated by the ability of humans and language models to accurately describe those features (Bricken et al., 2023; Cunningham et al., 2024; Rajamanoharan et al., 2024b). In this evaluation, humans and language models generate feature descriptions based on token sequences and their corresponding activations. The accuracy of these descriptions is then evaluated by predicting feature activations on unseen tokens. The correlation between predicted and true activations, typically quantified using the Pearson correlation coefficient, is used as a measure of description accuracy. However, recent work has critiqued this method, suggesting that even highly accurate feature descriptions may not faithfully represent the model being explained (Huang et al., 2023).

Alternative SAE evaluations analyze the sparsity of SAE outputs through the L0 norm, the presence of consistently inactive features, and 'loss recovered'. Loss recovered refers to the discrepancy in model loss between zero ablation of a layer and the insertion of an SAE output as if it were the activations of that model. The motivation for loss recovered is that it more directly approximates the information preserved by the SAE output, as this may not be accurately measured by the reconstruction loss.

## 2.3 SEMI-SUPERVISED LEARNING WITH MULTIPLE MODELS

Semi-supervised learning with multiple models involves training several models on both human-labeled and model-labeled data. Co-training, a semi-supervised technique, uses two distinct and conditionally independent views of the same data to iteratively improve the performance of two classifiers (Blum and Mitchell, 1998; Zhou and Li, 2005). This approach aims to maximize agreement between the classifiers and has been shown to improve their accuracy (Nigam et al., 2000). Similar techniques have been applied in deep learning for tasks such as machine translation (Xia et al., 2016) and image recognition (Qiao et al., 2018).

'Mutual learning' and 'co-teaching' have been used to describe techniques where student models trained in parallel teach each other by minimizing the divergence between their predictions. These methods have shown superior performance for the size of the student model compared to distillations of larger models (Zhang et al., 2017; Tarvainen and Valpola, 2017; Han et al., 2018; Ke et al., 2019; Wu and Xia, 2019). Related techniques include temporal ensembling, which improves model robustness by averaging predictions over multiple training epochs (Laine and Aila, 2016), and fraternal dropout, which encourages models trained in parallel to make similar predictions for the same data points, serving as a method of regularization to prevent overfitting (Żołna et al., 2017).

Our work builds on the semi-supervised learning literature. However, our motivation differs from the motivation for most of these techniques, which often relates to a lack of training data rather than learning features not in the input space.

## 3 EXPERIMENTS WITH SYNTHETIC DATA

Following Sharkey et al. (2022), we generate a synthetic dataset of vectors that represent more features than their dimensionality, allowing a direct measurement of how well an SAE learns the features of inputs with superposition. This simplifies our analysis by mitigating the problem of imperfect SAE evaluations (discussed in Section 2.2) as we can then directly compare the SAE and input features, but is only applicable when the input features are known, which is not typical for real-world data.

### 3.1 GENERATING SYNTHETIC DATA

We aim to create a synthetic dataset of vectors similar to activation vectors sampled from a neural network, but where the features of the input represented by the network are known and can be compared with the features learned by the SAE. These vectors should have similar properties to neural network activations, such as representing superposed features, and having correlations in the activation of features, but simultaneously be learnable by SAEs.

To do so, we generate a dataset $\mathcal{D} = \{\mathbf{x}^{(1)}, \mathbf{x}^{(2)}, \ldots, \mathbf{x}^{(N)}\}$ of vectors $\mathbf{x}^{(i)} \in \mathbb{R}^d$. Each vector represents the activation of $G$ features in $d$ dimensions, where $G > d$, is intended to mimic feature superposition in neural networks. We define the feature matrix $\mathbf{F} \in \mathbb{R}^{d \times G}$, where each element is sampled from a standard normal distribution:

$$F_{ij} \sim \mathcal{N}(0,1) \quad \forall i \in 1, \ldots, d, j \in 1, \ldots, G$$

We assign probabilities to each feature in $\mathbf{F}$ based on its index through exponential decay:

$$p_j = \frac{\lambda^j}{\sum_{k=1}^{G} \lambda^k} \quad \forall j \in 1, \ldots, G$$

where $\lambda \in (0, 1)$ is the decay rate hyperparameter, and is a specified constant. By raising $\lambda$ to the power of the index of the feature, we increase the decay for that feature such that the probability of a feature's activation decreases exponentially with its index.

To introduce correlations in the activations of features, we partition the features into $E$ groups of equal size $S = \frac{G}{E}$. Let $\mathcal{G}_e$ be the set of indices for features in group $e$:

$$\mathcal{G}_e = \{(e-1)S + 1, \ldots, eS\} \quad \forall e \in 1, \ldots, E$$

To construct each data point $\mathbf{x}^{(i)}$, we randomly select an active group $e_i \in 1, \ldots, E$ and choose $K$ active features within that group according to the probability $p_j$ of a feature. We denote this set $\mathcal{A}_i = \{\mathbf{f}_{1,i}, \ldots, \mathbf{f}_{j,i}, \ldots, \mathbf{f}_{K,i}\}$, where $\mathcal{A}_i \subset \mathcal{G}e_i$. Finally, we sample a sparse feature coefficients $a_{ij}$ for each feature in each sample according to:

$$a_{ij} = \begin{cases} u_{ij} & \text{if } j \in \mathcal{A}_i \\ 0 & \text{otherwise} \end{cases}$$

where $u_{ij} \sim \mathcal{U}(0,1)$ are the non-zero coefficients. $\mathcal{D}$ is created by linearly combining the ground truth features using the sparse feature coefficients:

$$\mathbf{x}^{(i)} = \sum_{j=1}^{G} a_{ij}\mathbf{f}_j$$

where $\mathbf{f}_j$ is the $j$-th column of the ground truth feature matrix $\mathbf{F}$.

### 3.2 TRAINING SPARSE AUTOENCODERS WITH MUTUAL REGULARIZATION ON SYNTHETIC DATA

We train SAEs with and without MFR on the synthetic dataset $\mathcal{D}$, generated with the parameters $G = 512$, $d = 256$, $E = 12$, $K = 3$ and $\lambda = 0.99$ in accordance with Section 3.1 (Figure 7). Samples in $\mathcal{D}$ are then comprised of 512 features represented in 256 dimensions with 36 active features, imitating feature superposition in neural networks. We train on 100 million unique examples.

For the MFR SAEs, we train two SAEs in parallel. A complete description of MFR is given in Section 3.3. All SAEs trained in this section have a hidden size of $512$, equalling the input feature count.

We train with a learning rate of $0.01$ with AdamW, and a batch size of $10,000$. On a 40GB A100 GPU, an SAE with these hyperparameters trains in approximately six minutes. 'Baseline' SAEs are trained only to minimize reconstruction loss, with sparsity enforced through the TopK activation function on the hidden state. When comparing baseline SAEs with MFR, we maintain an identical architecture and hyperparameter selection, excluding details specific to MFR. We use the exact value of total active features, 36, for the value of k in the TopK activation function for both

## 3.3 Analysis

We hypothesize that features learned by multiple SAEs trained on the same data will tend to correlate more strongly with a feature of the input than a feature learned by only one SAE. To test this hypothesis, we analyze the decoder weight matrices of two baseline SAEs, and compare them with the feature matrix $\mathbf{F}$ for their training dataset $\mathcal{D}$.

Let $\mathbf{W}'^{(1)}$ and $\mathbf{W}'^{(2)}$ represent the decoder weight matrices from two SAEs. For each feature $\mathbf{w}_i^{(1)}$ in $\mathbf{W}'^{(1)}$, we find the corresponding feature $\mathbf{w}_{j*}^{(2)}$ in $\mathbf{W}'^{(2)}$ that maximizes cosine similarity:

$$j^* = \arg\max_j \cos\left(\mathbf{w}_i^{(1)}, \mathbf{w}_j^{(2)}\right).$$

Likewise, for each $\mathbf{w}_j^{(2)}$ in $\mathbf{W}'^{(2)}$, we find the most similar feature $\mathbf{w}_{i*}^{(1)}$ in $\mathbf{W}'^{(1)}$ using the same cosine similarity maximization, resulting in pairs of features between the two SAEs that are most similar. To ensure a one-to-one correspondence of features, we use the Hungarian algorithm to assign the pairs.

We again use the same cosine similarity maximization, this time between $\mathbf{W}'^{(1)}$ and $\mathbf{F}$, as well as $\mathbf{W}'^{(2)}$ and $\mathbf{F}$, finding pairs of features between a decoder and feature matrix that are most similar. We plot these similarities for all features in $\mathbf{W}'^{(1)}$ and $\mathbf{W}'^{(2)}$ in Figure 2, illustrating the positive relationship (correlation coefficient = 0.625) between feature similarity across SAEs, and feature similarity with the input features.

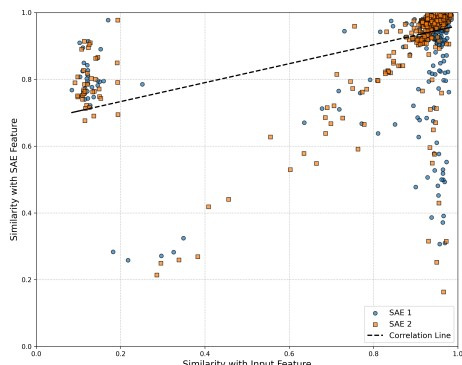

Figure 2: The relationship between feature similarity across SAEs, and feature similarity with the input features for two baseline SAEs.

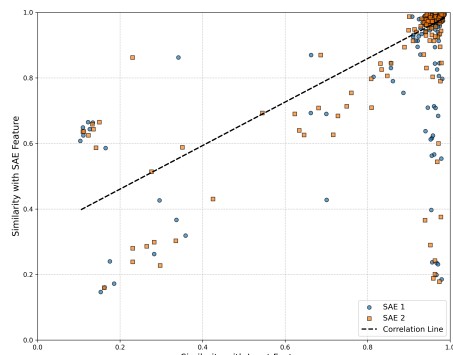

Figure 3: The relationship between feature similarity across SAEs, and feature similarity with the input features for two SAEs with conditionally reinitialized weights.

This correlation is weakened by a cluster of features with high similarity across SAEs, but low similarity with $\mathbf{F}$, potentially harming SAE performance due to the lack of similar input features for features in that cluster. We found that features in this cluster were significantly less likely to be active after the TopK activation function (Figure 4). By avoiding learning this cluster of features, we could improve SAE performance, as it comprises many of the features uncorrelated with those in $\mathbf{F}$. Additionally, it could increase the correlation between feature similarity across SAEs and feature

similarity with features in **F**, potentially allowing for further improvements in SAE performance by encouraging SAEs to learn features present in both their decoder, and the decoders of other SAEs trained on the same data.

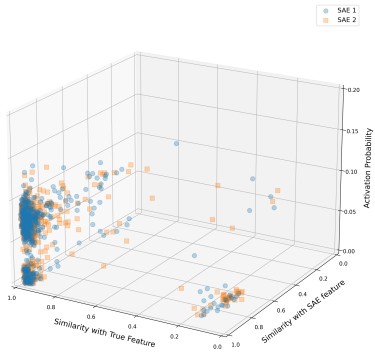 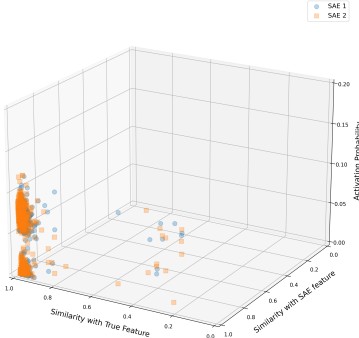

Figure 4: The relationship between feature similarity across SAEs, feature similarity with the input features and the likelihood a feature is active after the TopK activation function on the hidden representation for two baseline SAEs.

Figure 5: The relationship between feature similarity across SAEs, feature similarity with the input features and the likelihood a feature is active after the TopK activation function on the hidden representation for two SAEs trained with MFR.

Across multiple training runs, a subset would have a reduction in the size of this cluster, resulting in SAEs learning features that correlate more strongly with features in **F** (Figure 8). This variation was binary: either the cluster would be larger, at approximately 15 features per SAE, or smaller, at approximately 5. We did not observe other variations. As no hyperparameters were modified, we hypothesize that this is caused by differences in the random weight initializations, and found that we could reliably detect these superior weight initializations by the presence of features consistently not active after the TopK activation function, often in the first 100 training steps (Figure 6).

By reinitializing the SAE weights if a measure of these inactive features exceeded a threshold, we consistently find initializations that do not result in that cluster of features being learned. Doing so strengthens the correlation between the similarity of features learned across SAEs, and the similarity of features learned by the SAEs with **F** (correlation coefficient = 0.625 increased to 0.668) (Figure 3).

We found that the particular metric used to decide whether to reinitialization the SAE weights did not effect performance, as the behavior of initializations with smaller clusters of these features uncorrelated with features in the input feature matrix were easily identified by all metrics tested that measure feature inactivity after the TopK activation function. We give an example in Figure 6, plotting the deviation of features from the mean activation probability of a feature, calculated as the value of $k$ used for the TopK activation function divided by the decoder size.

Finally, to incentivize features present in the decoders of other SAEs trained on the same data, we add an auxiliary penalty to the SAE loss function. We define this auxiliary penalty as

$$\frac{\alpha}{\binom{N}{2}} \sum_{i=1}^{N-1} \sum_{j=i+1}^{N} \left(1 - \text{MMCS}(\mathbf{W}^{(i)}, \mathbf{W}^{(j)})\right)$$

where $\alpha$ is a constant that weights the penalty, $N$ is the number of SAEs, and MMCS is a function that returns the mean of the max cosine similarity pairs across the weight matrices $\mathbf{W}^{(i)}$ and $\mathbf{W}^{(j)}$. We calculate MMCS as

$$\text{MMCS}(\mathbf{W}^{(i)}, \mathbf{W}^{(j)}) = \frac{1}{|\mathbf{W}^{(i)}|} \sum_{w_i \in \mathbf{W}^{(i)}} \max_{w_j \in \mathbf{W}^{(j)}} \text{CosineSim}(w_i, w_j).$$

We name the combined use of our reinitialization method and auxiliary penalty MFR. We find that MFR results in SAEs recovering more of **F** (Figure 8), and that SAEs trained with MFR did not have the cluster of features with high similarity across SAEs, but low similarity with **F** '(Figure 5).

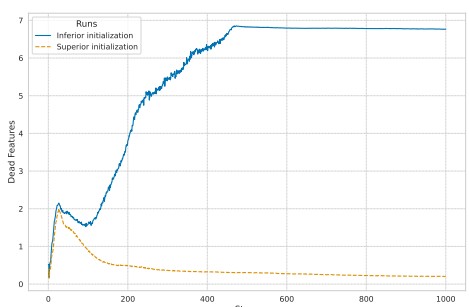
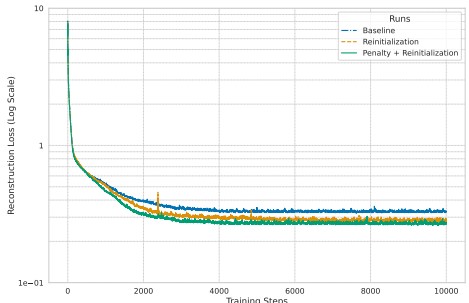

Figure 6: We plot $\frac{1}{N}\sum_{i=1}^{N}\left(\frac{|\mathbf{W}x_i-(k/N)|}{k/N}\right)$, where $N$ is the neuron count of $\mathbf{W}$ and $k$ is the number of active neurons in the hidden layer after $\sigma_k$. $k/N$ is then the frequency each feature would be active if all features were equally likely to activate. Hyperparameters were identical across runs.

Figure 7: The reconstruction loss of baseline and MFR SAEs. The reconstruction loss scale is logarithmic to better display the separation of reconstruction losses. The relative difference in the 'Reinitialization' and 'Penalty + Reinitialization' reconstruction losses at the final training step is 2.4%.

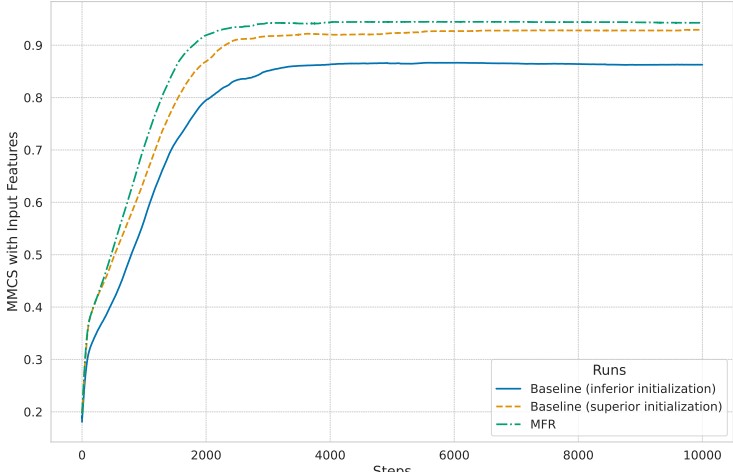

Figure 8: MMCS of the decoder weights with the input feature matrix of a baseline SAE, and two SAEs trained with MFR. The MFR and 'superior initialization' SAEs are reinitialized if $\frac{1}{N}\sum_{i=1}^{N}\left(\frac{|\mathbf{W}x_i-(k/N)|}{k/N}\right)=1$, which serves as a threshold of feature inactivity. For the MFR SAE, the constant $\alpha$ that weights the auxiliary penalty is set to 3.

## 4 Scaling mutual regularization

In this section we scale MFR to larger models and real-world data. We train SAEs with MFR to reconstruction activations sampled from GPT-2 Small, or to reconstruct EEG data, showing improved performance compared to baselines. We choose these tasks because they demonstrate the results in Section 3 generalize to natural data from a neural network, and to a non-interpretability task: denoising.

### 4.1 GPT-2 Small

We train five baseline and five MFR SAEs for 2,000,000 tokens on the first layer MLP outputs of GPT-2 Small, constraining the active neurons in a hidden layer of size 3072 to 6, 12, 18, 24 and 30 respectively. We use a batch size of 500, and a learning rate of 0.001 with AdamW. On a single V100, this takes approximately 2 hours to train both baseline and MFR SAEs.

For the MFR SAEs, we set the coefficient that weights the auxiliary penalty $\alpha$ such that the initial reconstruction loss and auxiliary penalty are equivalent, and use a 100 training step cosine warmup for the penalty. The penalty is applied to all five SAEs trained with MFR, such that they are all encouraged to learn similar features in training. We found that the warmup could prevent features becoming too similar early in training, and would allow setting $\alpha$ large enough to cause convergence later in training without increasing the reconstruction loss.

The three SAEs with the smallest values of $k$ (6, 12, 18) achieved superior reconstruction loss using MFR (Figure 9). For the two remaining SAEs ($k = 24$ and $k = 32$), we found equivalent or inferior reconstruction loss. Over the five SAEs, we found a mean reduction in the reconstruction loss of 5.66%. The most significant improvement was in the $k = 6$ SAE, with a reduction of 21.21%, and the most significant degradation was in the $k = 30$ SAE, with an increase of 7.89% (Table 1).

| k | MFR | Baseline | Relative Difference |
|---|---|---|---|
| 6 | **0.00132** | 0.00160 | -21.21% |
| 12 | **0.00121** | 0.00135 | -11.57% |
| 18 | **0.00116** | 0.00122 | -5.17% |
| 24 | 0.00114 | **0.00112** | 1.75% |
| 30 | 0.00114 | **0.00105** | 7.89% |

Table 1: Comparison of the final reconstruction accuracy of SAEs trained with and without MFR.

MFR consistently results in superior loss recovered compared to baselines (Figure 9). For this metric we extract the layer 0 MLP outputs of GPT-2 Small and reconstruct them using SAEs. We then insert the SAE outputs as though they were the MLP layer outputs, and measure the cross-entropy loss of the model on 10,000 randomly selected sequences from OpenWebText (Gao et al., 2020). We find a mean improvement of 5.45% in the MFR SAEs, a maximum improvement of 8.58%, and a minimum improvement of 3.51% (Table 2). With no modifications, GPT-2 Small's cross-entropy loss on this dataset is 3.12, and 132.27 with the first MLP layer zero ablated.

| k | MFR | Baseline | Relative Difference |
|---|---|---|---|
| 6 | **10.121** | 10.798 | -6.27% |
| 12 | **9.782** | 10.300 | -5.03% |
| 18 | **9.367** | 9.742 | -3.85% |
| 24 | **10.136** | 10.505 | -3.51% |
| 30 | **9.624** | 10.527 | -8.58% |

Table 2: The cross-entropy loss of GPT-2 Small on a subset of OpenWebText2 with SAE outputs inserted as MLP outputs.

We believe these results suggest MFR causes SAEs to learn more information about the features that underly their training dataset. Specifically, the reduced loss recovered indicates the SAEs preserve more information about their inputs, and the improvements in reconstruction loss show more accurate reconstructions in terms of Euclidean distance to the input, but that this depends on the value of $k$ in the TopK activation function relative to the other SAEs being trained.

### 4.2 ELECTROENCEPHALOGRAPHY DATA

We train five baseline and five MFR SAEs for 3,500,000 tokens at a learning rate of 0.001 with AdamW on vectorized EEG data from the TUH EEG Corpus (Obeid and Picone, 2016). We use a hidden size of 4096 and values of 12, 24, 36, 48 and 60 for the TopK activation function. We preprocess the EEG data with a low-cut frequency of 0.5Hz, a high-cut frequency of 45Hz and a filter order of 5. $\alpha$ is set to equal the initial reconstruction loss, and we use a cosine warmup of 100 steps. The auxiliary MFR penalty considers all five MFR SAEs. On a single V100 GPU, with the above hyperparameters, the five baseline and MFR SAEs train in one hour with a training batch size of 1024.

We train on EEG data to show that MFR can be applied to SAEs trained on non-neural network data. SAEs have been applied to EEG denoising in the past (Qiu et al., 2018; Li et al., 2022), and in

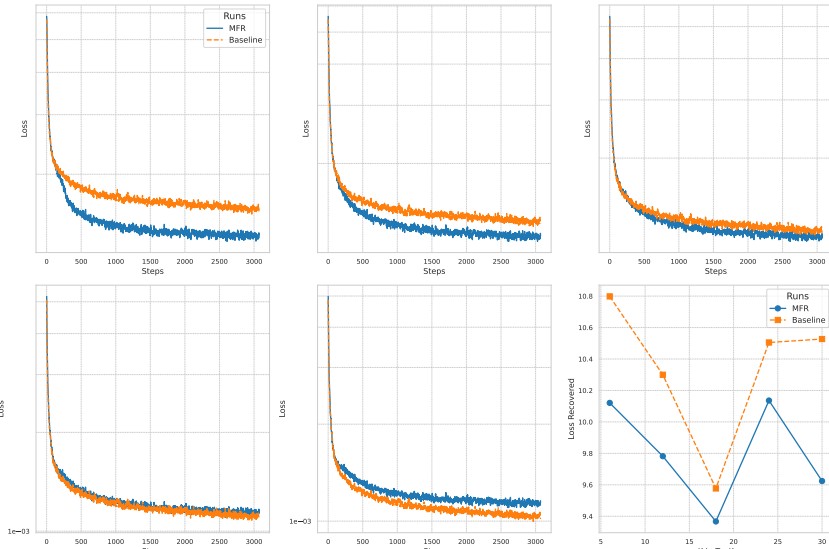

Figure 9: The reconstruction loss and loss recovered of various SAEs trained on activations from the first MLP layer of GPT-2 Small. We plot the reconstruction loss on a logarithmic scale.

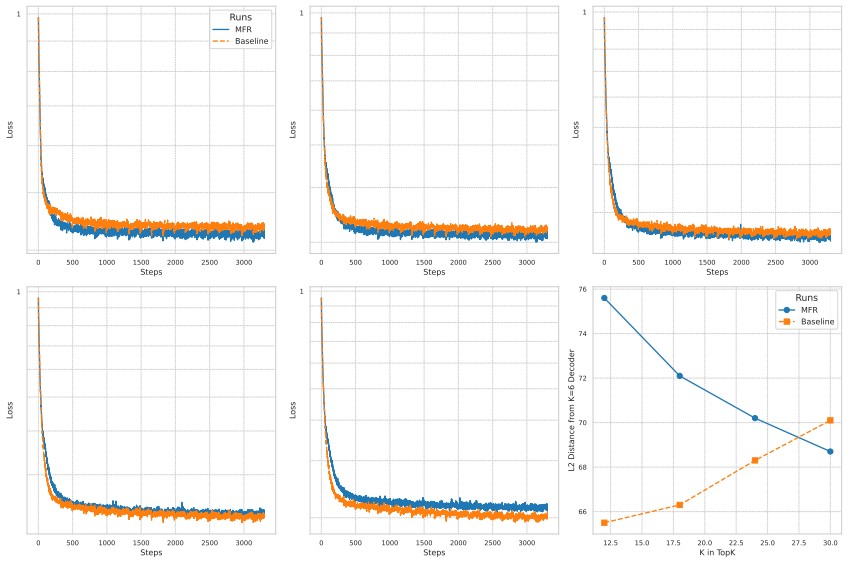

Figure 10: The reconstruction accuracy and loss recovered of various SAEs trained on vectorized EEG data from the TUH EEG Corpus.

both finding more interpretable representations of neural network activations and denoising accurate feature learning is beneficial, so plausibly MFR is useful for denoising EEG data.

For the reconstruction loss, we find a mean improvement of 1.8%, a maximum improvement of 6.67%, and a maximum degradation of 4.04% (Figure 10). The benefits of MFR on this dataset are reduced significantly from Section 4.1. We hypothesize that this is because MFR is designed to encourage SAEs to learn accurate representations of input features in which features are represented with superposition in the training data. Although there is evidence that features are superposed in neural network activations (Elhage et al., 2022; Jermyn et al., 2022), the same evidence is not present for EEG data.

| k | MFR | Baseline | Relative Difference |
|---|------|----------|---------------------|
| 12 | **0.42** | 0.44 | -4.76% |
| 24 | **0.30** | 0.32 | -6.67% |
| 36 | **0.26** | 0.27 | -3.85% |
| 48 | 0.23 | **0.23** | 1.80% |
| 60 | 0.21 | **0.20** | 4.04% |

Table 3: The reconstruction loss of SAEs trained with and without MFR on vectorized EEG data from the TUH EEG Corpus, and the L2 distance of the decoder weights of those SAEs from the decoder weights of the $k$=6 SAE. We plot of the reconstruction loss on a logarithmic scale.

### 4.3 WEIGHT ANALYSIS

One concern with MFR may be that in encouraging the SAE decoder features to be similar, the decoder weight matrices end up more similar than without MFR. This could be problematic, as SAEs with lower values of $k$ for the TopK activation that were trained with MFR alongside SAEs with higher values of $k$ could end up less sparse by becoming more similar to the SAEs with higher values of $k$. To test this, we measure the L2 distance of the decoder weight matrices for the baseline and MFR SAEs trained in Section 10.

We find that SAEs trained with MFR tend to be more different in terms of the L2 distance, but that as $k$ increases they trend toward lower L2 distances (Figure 10). This is in contrast to the baseline SAEs, which are more similar at smaller values of $k$, but trend towards larger L2 distances as $k$ increases. At the values of $k$ we train at, we do not consider this problematic, as the L2 distances suggest the decoder weight matrices are more different on average rather than less.

## 5 CONCLUSION

We proposed a method for training SAEs designed to recover more features of the input. We first establish a motivating hypothesis for MFR, that feature similarity across SAEs is correlated with feature similarity to the input features, showing that this hypothesis is true for SAEs trained on synthetic data, and that MFR improves the fraction of features of the input recovered (Section 3). We then scale MFR to both language model activations and a realistic denoising task, and show that it improves SAE performance on key metrics at scale (Section 4). We believe our method encourages SAEs to learn more features of the input, increasing their usefulness for interpretability.

### LIMITATIONS

Although we show improved performance of SAEs with MFR, this comes at a relative increase in computational cost, as the auxiliary penalty used in MFR requires training additional SAEs. As all of our experiments are easily completed on a single GPU, this is not problematic in our work. However, larger models can require SAEs with very large hidden dimensions, making this cost unmanageable if SAEs need to be trained for many layers. Training a multiple of the SAEs that would need to be trained without the auxiliary penalty may not be justifiable depending on the scale of the experimentation.

Despite the increased computational requirements, we believe that the auxiliary penalty is still valuable due to the small computational budget for SAEs relative to the models they are trained on. For smaller models (where the cost of training SAEs is less significant), it may be worth the increase in training compute for more accurate SAEs. For example, in the case of GPT-2 Small, the additional compute may not be of concern, as training is manageable on a single GPU or a small cluster making the additional information about the input features worth the additional compute.

We hope that future work will investigate efficient mutual learning-based approaches for SAEs that can benefit from the positive relationship between feature similarity across SAEs, and feature similarity with the input features without requiring more compute.

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
