# OpenReview forum: "Enhancing Neural Network Interpretability with Feature-Aligned Sparse Autoencoders"
_ICLR.cc/2025/Conference — Submitted to ICLR 2025_

### Official Review · Reviewer_3gqg · 2024-10-28

**Soundness:** 2
**Presentation:** 1
**Contribution:** 2
**Rating:** 3
**Confidence:** 3

**Summary:**

This paper introduces a method of enhancing the feature interpretability of a learned neural network via sparse auto-encoders. Specifically, a mutual feature regularization technique has been proposed by correlating multiple SAEs. The experiments were conducted on a simulated dataset and two real-world datasets.

**Strengths:**

The paper addresses the important issue of interpreting an internal representation of a trained network.

Three datasets, one simulated and two real, were used to support the validity of the proposed method.

**Weaknesses:**

The paper needs to reorganize its structure to describe its ideas and contributions better. In its current form, it isn't easy to understand.
The technical contribution is marginal and is not well supported by the experiments.

This paper is about enhancing the interpretability of an internal representation of a neural network. However, there are quantitative values compared with baseline models, but there are no interpretation-related results or explanations in the experiments.

When training SAEs, the hidden size plays an important role, which may cause the feature superposition or feature distribution to occur.
In this respect, there should be a comparison among varying hidden sizes in SAEs.

The current experimental results do not support many hypothetical statements well in Section 3.3. The validity of their arguments needs rigorous efforts.

Page 6, line 307: A statistical significance test is needed to justify the increase in correlation with reinitialization better.

**Questions:**

Page 5, lines 225-226: “We see the exact value of total active features, 36, ~~.” What do the authors mean by the exact value, and where does the value 36 come from?

How exactly does the proposed MFR is defined? It is not explicitly entailed.

---

> ### Author Response · Authors · 2024-11-21
>
> Thank you for your insightful review.
>
> > “This paper is about enhancing the interpretability of an internal representation of a neural network. However, there are quantitative values compared with baseline models, but there are no interpretation-related results or explanations in the experiments.”
>
> We acknowledge the absence of a verification that SAEs trained with MFR permit better downstream analysis. We would commit to this experiment for a camera-ready revision:
>
> 1. Collect data on the activation of SAE features on tokens for the GPT-2 SAEs
> 2. Task humans with explaining the features based on the data from (1)
> 3. Task humans with predicting the activations of SAE features on unseen tokens given the explanations in (2), where more accurate predictions of the activations on unseen tokens indicate more interpretable features
>
> At the reviewers request we could perform this experiment using a language model such as in [Bills et al.](https://openaipublic.blob.core.windows.net/neuron-explainer/paper/index.html), however as mentioned in Section 2.2, recent work has brought this evaluation into question, showing a lack of causal relevance of language model feature descriptions. Ultimately we feel skeptical that there is a good method of evaluation for the interpretability of SAE features.
>
> > “When training SAEs, the hidden size plays an important role, which may cause the feature superposition or feature distribution to occur. In this respect, there should be a comparison among varying hidden sizes in SAEs.”
>
> For the SAEs we trained on the synthetic dataset, we already know the true input feature count, so we can optimally set the SAE hidden size without needing to search over hidden sizes.
>
> > “The current experimental results do not support many hypothetical statements well in Section 3.3. The validity of their arguments needs rigorous efforts.”
>
> We politely request which statements the reviewer feels are not well supported. We feel that we have supported the primary claims of the section. We view these as:
> 1. Showing the correlation between a feature being learned by multiple SAEs and it being an input feature (we show this correlation in Figure 2).
> 2. Features that are similar across SAEs but not with the input features can often be detected by how active they are (Figure 4).
> 3. By avoiding the features mentioned in point (2) and encouraging similarity in the decoder features we learn more of the input features (Figure 8).
>
> Per your questions:
> 1. This is the number of active features per data point we specified when generating the synthetic dataset. Because the data is synthetic, we are able to control this and manually determine the active features per data point.
> 2. MFR is the combination of our reinitialization technique and auxiliary penalty. We define the auxiliary penalty from lines 312-314, and detail the reinitialization from lines 300-309 of the revised PDF.

---

> > ### Comment · Reviewer_3gqg · 2024-11-27
> >
> > Thank you for answering my concerns. However, the interpretability-related issues remain unresolved. In particular, since the synthetic dataset's true feature count is known in advance, comparing and analyzing the results according to the varying hidden sizes is a nice situation.

---

> ### Author Response · Authors · 2024-11-25
>
> Thank you for your time in reviewing our submission. As the rebuttal period is nearly over, we wanted to kindly follow up to see if our rebuttal addressed some of the points in your review, or if there were any further questions we could answer.

---

### Official Review · Reviewer_6tre · 2024-11-02

**Soundness:** 2
**Presentation:** 2
**Contribution:** 2
**Rating:** 5
**Confidence:** 4

**Summary:**

The paper investigates a mutual alignment regularization to train a set of sparse auto-encoders (SAE) in the context enhancing efficiency and interpretability. It does so by penalizing the elements in one dictionary (decoder basis) from not being matched with at least one  element in another dictionary. The method is validated on synthetic data created with a sparse, group-structured model. Only one group is active and $K$ elements in the group are active, based on underlying rates. The method is also tested on a small trained transformer GPT2 small and finally on EEG data. The results show that with fewer active components the regularized networks outperform the unregularized network in terms of reconstruction cost, and for the GPT2 corpus in terms of the cross-entropy loss.

**Strengths:**

Directly ensuring robustness of SAE through an alignment loss appears original and seems to provide some gains. It addresses a problem in the use of SAE-based interpretation of networks.

**Weaknesses:**

Based on the premise I was expecting more resulting showing improved interpretation. While quantitative results are given, the significance of them is questionable.

The paper's organization could be improved by stating initially the formulation. Figure 1 does not convey much information.  It says on line 81 that "auxiliary penalty calculated using the similarity of the SAE weights" is used, but it is not introduced until Section 3.3 titled "Analysis" there is finally a formulation (in numbered equations) at the bottom of page 6.  Jumping directly to the description of the synthetic data seems out of place.

The formulation of the auxiliary penalty/regularization itself is cryptic. It appears to involve asymmetries (perhaps due to the choice of cardinalities) that are not motivated.  If this is used to regularize all networks, due to the asymmetry, it seems that the index of the network adjusts how it is applied.

The choice of number of SAEs in the regularization isn't precisely given. I take it that the SAEs trained at different levels cardinality (TopK) are co-learned, but this isn't stated explicitly.

Some of the statements early on are not clear (noted in question/comment set 1 below).

The use of EEG data is not well motivated nor are the details of the data well explained. If there was a downstream task like in GPT2 case that would be more interesting.

**Questions:**

Question/comment Set 1:
-"features that are not features of the input"  Rather directly correlate with known features of the input.
- The expression  "features that correlate with features of the input" should be clarified with the fact that the latter corresponds to human interpretability.
-I assume reconstruction loss in the abstract is at the same sparsity level.
-Are polytopes "fundamental units of neural networks" like unit activations?
-The premise of "studying an SAE would not be guaranteed to reveal information about the neural network that SAE was trained on activations from" is strange. Just because the SAE activations aren't as interpretable they are still a function of the activations of the neuron network. Of course by the information processing inequality they can contain less information (or possibly zero if they don't pass anything). However, I do agree with the premise that "studying SAEs would [not always] be more fruitful than studying the raw activations directly".
- It should be made clear that the 'model loss' in the definition of 'loss recovered' refers to a task loss, perhaps the training loss, but could also be a loss function used in validation. References for this definition should be given.



Comment 2:
While not studied formally the significance of the problem reminds me of work studying whether LASSO is a consistent estimator of the true support. The authors considered whether random initialization of the weights is necessary if bootstrap sampling could be used to create different training samples for the same data. This is explored in Francis Bach's work on bootstrap LASSO, which uses the intersection of the support of sparse linear models to find a reliable support. It would be interesting if the proposed regularization works from simply different sampling of the same data without random initialization of the weights. In this case the weights may correspond without the quadratic cost.


Comment 3:
The Hungarian algorithm is used to assign pairs to maximize correspondence with results in Figures 2, 3, 4 and 5; however, the linear assignment problem assumes one-to-one so if there is duplication of a neuron/dictionary element it will lead other pairs to be improperly matched.  So it is not clear to me that when the result shows low similarity, which is interpreted there wasn't a correspondence, it is in fact the case that the assignment didn't correspond.  Partial optimal transport methods that don't require the marginals to be matched exactly can be used to give a more robust matching.

Comment 4:
The synthetic data has some arbitrary choices of simulation (groups all being the same size). Furthermore, after $p_j$ is chosen as active, wouldn't it make sense to bound that sparse features coefficients away from 0? Choosing uniform over [a,1] with a>0 or some other distribution.

Comment 5:
It is not clear to me why a cluster of features that are similar between SAEs but are to the input feature matter if they are not active in the TopK activation? While I agree that reinitializing those features (which to my understanding is common for SAEs) can make the network more efficient.

Comment 6: Asymmetry in penalty across models. See comment in Weaknesses.

Comment 7:
In Figure 6, the definition of the average of the absolute value is not clear. Why would the activation be k/N in the top-K. The top-K doesn't ensure sums to 1, but rather that it is above the threshold. Otherwise the metric is not meaningful. Its not clear if the reset threshold in caption Figure 8 makes sense.

Question set 8:
Line 363, it is not clear what the initial reconstruction loss was? Is it the first batch, the first epoch? What does the "100 training step cosine warmup for the penalty" mean?

Comment 9:
The lower loss on the task loss in Table 2 is interesting. Especially since the lowest loss is achieved at at k=18 which isn't the lowest reconstruction loss. However, it is hard to tell if this difference is meaningful as the loss is much higher than the no modification loss, while much lower than the zeroing out ablation.

Question 10:
For the EEG data it is not clear what is meant by vectorized. Is it multichannel windows that is vectorized? If so what is the size of the windows, sampling rate, and number of channels?

Minor points:
-Lines 97–100, need parenthetical references.  Also on lines 126-127, lines 132-133, 138–139 and elsewhere.
-Line 129 "to optimizing"
-Lines 140–141 is not a complete sentence.
-Figure 10 doesn't convey any information to the reader (especially without log y-axis).

---

> ### Author Response · Authors · 2024-11-21
>
> Thank you for your insightful review.
>
> 1. By “features that are not features of the input”, we do not mean with respect to human interpretability, we mean that the features are fundamentally different from all features of the input. We do not claim that polytopes are the fundamental units of neural networks. Previous literature has chosen to study a particular division of the neural network (e.g., polytopes) and treat the neural network as being composed of that unit, which is what the quote in question is describing.
> 2. We are concerned that bounding the sparse feature coefficients away from zero could reduce how realistic the sparsity is, and bias the dataset toward over-representing less significant features. We feel that doing this changes the task significantly in a way that makes it less suitable for a sparse autoencoder.
> 3. It is concerning to us to have a cluster of features similar between sparse autoencoders that was not similar to the input features even if those features are inactive because (1) These features are occupying space in the decoder weights that could be replaced with features more similar to the input features, and (2) It makes the expected gains of encouraging similarity in the decoder weight matrices of multiple sparse autoencoders less, as the correlation between the similarity with the input features and with the features of other sparse autoencoders weaker.
> 4. We suggest that the asymmetry of the auxiliary penalty is not inherently problematic. At the reviewers request we could experiment with a symmetric penalty (e.g. by modifying the MMCS calculation to be an averaged sum of the MMCS of the first decoder features to the second, and then the second decoder features to the first) in a camera-ready revision.
> 5. We believe the threshold in Figure 8 is appropriate because it refers to the probability a feature is in the top-K, not to its activation value. If all features were equally likely to be in the top-K, then they would have an activation probability of k/N, where N is the number of features.
> 6. The initial reconstruction loss is the reconstruction loss after the first batch. The "100 training step cosine warmup for the penalty" refers to a cosine warmup on the coefficient for the auxiliary penalty over the first 100  batches.
> 7. Because the reconstruction loss is an imperfect measure of how much information the decoder weights preserve about the input features, we expect that the reconstruction losses may not match up perfectly with the performance on downstream applications of the autoencoders.
> 8. Yes, we vectorize multichannel windows. Each window represents one second of EEG data, and so represents 250 (the sampling rate) samples per channel per window.
> 9. Please see the updated PDF with a revised Figure 10 that should be much more readable.

---

> > ### Comment · Reviewer_6tre · 2024-11-23
> >
> > Thank you for your answer and revision of Figure 10. I think the clarifications help, but do not cover all my concerns and the key weaknesses remain. In particular, I'm not sure about addressing the lack of testing "Enhancing Neural Network Interpretability" by additional human labeling experiments that have yet to be performed ("We would commit to this experiment for a camera-ready revision:").
> >
> > I think the idea of co-learning is interesting and is very well suited to sparse auto-encoders. So I'm raising my score marginally to account for the case that the proposed co-learning improves the consistency of the SAEs.

---

> > > ### Author Response · Authors · 2024-11-25
> > >
> > > We are glad that we could provide some clarifications, and appreciate you revising your score in light of those. We are also glad you find the idea interesting and well suited to SAEs. Thank you again for your review and suggestions.

---

### Official Review · Reviewer_R6cT · 2024-11-03

**Soundness:** 4
**Presentation:** 3
**Contribution:** 2
**Rating:** 3
**Confidence:** 4

**Summary:**

The authors propose a regularization framework that both determines whether sparse auto-encoders need to be re-trained and enforces sparse autoencoders trained with different random initializations are learning the same features. The authors validate their results on a simulated dataset, and then apply their model on GPT-2 and EEG data.

**Strengths:**

This work is of high quality and largely original. Moreover, reading the paper is easy, and the authors do a great job building up their intuitions, related work, and simulation results to verify their model. Results are explained incredibly well, examples of this are Figures 2-6, and I think their method can be significant. I think everything up to the real world results are done incredibly well, and I think the authors have a good base for a strong paper.

**Weaknesses:**

Major weakness(es): \
As good as the paper is up to its real world results, I believe the real world results are not as convincing. One of the main issues with the real world results is that the authors do not provide any deeper insights about what their improvements tell us about the data they are analyzing. One main feature of their method in the title is to improve interpretability, but when they apply the model on real world data, they do not actually verify new interpretations, i.e. what does the author’s model tell us about EEG data that the baseline did not? How do the improvements in reconstruction accuracy translate into actionable or interesting results for both the GPT-2 and EEG results? In fact, in their introduction (Section 2.2) the authors mention in Lines 128-130 that “… the reconstruction loss itself is not sufficient to evaluate SAEs, …” which I tend to agree with, but Tables 1-3 and Figure 9&10 only evaluate exactly that (and cross-entropy for GPT-2, but in some sense this is a different measure of reconstruction accuracy: prediction preservation accuracy). I think the authors can make this paper very impactful and significant if they explain to the reader what the added improvements in reconstruction accuracy/cross entropy loss translate to in terms of interpretability, but at this moment the paper seems unfinished. I think the idea the authors propose is important, and I think that with substantial improvements in the regard that I have just described, I am open to significantly increasing my score.

Minor weakness(es):
1. Figure 8 does not show standard deviations, and the authors do not report significance tests for the numbers in their tables (1-3).
2. The authors show many training curves, I believe these should not be added in the main text unless they add something, moving them to the Appendix would provide the authors enough space to dig into interpretability results from their model.
3. The performance of the authors’ model is not better for 24 and 30, 48 and 60 for the GPT-2 and EEG results, respectively. The authors should explain better why this is the case, and potentially research why this phenomenon happens. Since just from a reconstruction error perspective, out of the models the authors have tested their baseline performs the best for both the GPT-2 and EEG data for k=30 and k=60, respectively. If the authors can show that smaller ks help with interpretability, then this is not perse an issue, but since most of the results currently do not actually dive into interpreting the results and mostly focus on reconstruction accuracy, it hurts the impact of the paper.
4. The authors cite two papers on Lines 466-467 that perform EEG denoising, but do not add these papers as baselines.
5. Some of the hyperparameters are not explained, e.g. a hidden size of 4096 and learning rate of 0.001 when training on the EEG data. To improve the confidence in their paper, I would suggest running atleast a small hyperparameter study using the validation set, and selecting the best model for the test set (this applies to all of the authors’ results, including the simulation results).

**Questions:**

Why did the authors decide to use EEG data, and specifically the TUH dataset? \
Line 427: Why is the author’s model only better in terms of reconstruction error for small k values? See table 1 and 3 \
Line 320/321: Do the authors return the mean of the max cosine similarity pairs based on the Hungarian algorithm output? \
Line 244: Why do the authors use cosine similarity maximization and not another distance metric? Since cosine similarity does not take the magnitude of the two matrices into account.

---

> ### Author Response · Authors · 2024-11-21
>
> Thank you for your review, we are glad you found the work of high quality.
>
> We acknowledge the absence of a verification that SAEs trained with MFR permit better downstream analysis. We would commit to this experiment for a camera-ready revision:
> 1. Collect data on the activation of SAE features on tokens for the GPT-2 SAEs
> 2. Task humans with explaining the features based on the data from (1)
> 3. Task humans with predicting the activations of SAE features on unseen tokens given the explanations in (2), where more accurate predictions of the activations on unseen tokens indicate more interpretable features
>
> At the reviewers request we could perform this experiment using a language model such as in [Bills et al.](https://openaipublic.blob.core.windows.net/neuron-explainer/paper/index.html), however as mentioned in Section 2.2, recent work has brought this evaluation into question, showing a lack of causal relevance of language model feature descriptions. Ultimately we feel skeptical that there is a good method of evaluation for the interpretability of SAE features.
>
> We acknowledge the minor weaknesses you mention, and would hope to uniformly address these in a camera ready version.
>
> For your questions:
> 1. We speculate that MFR is more useful for the more sparse autoencoders because they are more likely to learn features that are not features of the input. We suggest that the extreme sparsity constraint on these models could incentivize feature composition, or hinder feature learning for the baseline autoencoders. Because there is information transfer between all autoencoders, this could hinder larger autoencoders by encouraging them to learn the features from the extremely sparse autoencoders.
> 2. Yes.
> 3. We view cosine similarity not accounting for magnitude as beneficial. We are ultimately interested in the features represented by the decoder weights rather than measuring the general similarity of the weights, so relative orientation seems the most important property to measure.

---

> > ### Author Response · Authors · 2024-11-25
> >
> > We appreciate your effort and time in reviewing our submission. As the rebuttal period is in its final days, we wanted to kindly follow up to see if our rebuttal addressed some of the points in your review, or if there was any further clarification we could provide.

---

> ### Comment · Reviewer_R6cT · 2024-11-25
> **Rebuttal response**
>
> Dear authors, thank you for your rebuttal. I retain my score because I can't change my score based on an experiment that hasn't been done yet. I do want to encourage the authors to explore this line of research further and I think this work can be impactful and significant with more results that delve more into the sparse features of the EEG and GPT-2 data. A few recommendations I have for the authors are:
> - Visualize what each sparse feature corresponds to in the EEG signal, e.g. are these specific brain regions? Generally designing an experiment that tries to look at whether the sparse features for example reduce noise in the EEG signal or are more interpretable than ICA would improve the impact of the paper for EEG research.
> - Regarding my questions: adding your answers more clearly in the paper will increase the clarity of the paper, e.g. mention your assumption about cosine similarity when you introduce it as a metric (q3). Additionally, I would try to incorporate an experiment that verifies that "... MFR is more useful for the more sparse autoencoders because they are more likely to learn features that are not features of the input. We suggest that the extreme sparsity constraint on these models could incentivize feature composition, or hinder feature learning for the baseline autoencoders."
>
> I would be happy to give you feedback on the experiment you propose in your rebuttal, but I don't quite understand the experiment the way it is presented in the rebuttal. Could you expand on the experimental design?
>
> Thank you!

---

> > ### Author Response · Authors · 2024-11-30
> >
> > Thank you for the recommendations and again for your engagement throughout the review process.
> >
> > On consideration, there is an experiment that we believe would better assess if SAEs trained with MFR learn more features of the input when those features are not known than the experiment proposed in our initial response. In this experiment, we would train many probes on the activations of GPT2-Small, each to detect the activation of a particular feature of the input (this would involve estimating a large quantity of features learned by GPT2-Small). We would then train probes to detect the same features in the outputs of SAEs trained with and without MFR. If SAEs trained with MFR learn more features of the input, more of the probes should successfully train on the MFR SAE outputs than the baseline SAE outputs.
> >
> > This type of analysis has appeared previously in the literature (e.g., Section 4.2 of [Gao et al.](https://arxiv.org/pdf/2406.04093)). Your feedback on this experiment would be greatly appreciated.

---

> ### Comment · Reviewer_R6cT · 2024-12-03
> **Experimental design**
>
> Dear authors,
>
> Thank you for sending over the proposed experimental design.
> I think the experimental design sounds good, the authors could also use this approach to further argue that a smaller k (number of sparse latent dimensions in the autoencoder) is better (in the cited paper, Figure 6 shows that a higher number of sparse latent dimensions in the autoencoder is not necessarily better). However, I would additionally urge the authors to implement an experiment like Section 4.3 from the same paper as well. By showing how quantitative explanations differ across random initializations and are better with MFR the authors can even more clearly motivate their method. The authors could show a quick comparison either in the main text or Appendix across a few random initializations to show that differently initialized models can learn different sparse features, and then show how using MFR makes explanations more stable, i.e. similar across random initializations (I assume this is the case, but the authors could also explicitly show this), and more aligned with human interpretability.

---

### Official Review · Reviewer_S2HV · 2024-11-04

**Soundness:** 2
**Presentation:** 2
**Contribution:** 2
**Rating:** 5
**Confidence:** 3

**Summary:**

The manuscript introduces Mutual Feature Regularization (MFR), a technique to improve Sparse Autoencoder training by encouraging multiple SAEs trained in parallel to learn similar features. Using synthetic data with known features, the authors validated that MFR helps SAEs learn actual input features more effectively than SAE. Furthermore, the technique improves reconstruction loss by up to 21.21% on GPT-2 Small activations and 6.67% on EEG data.

**Strengths:**

- Authors performed experiments with synthetic, EEG, and GPT2 weights datasets.

**Weaknesses:**

- Tables 1, 2, and 3 do not report variability across folds or multiple runs.
- The regularization has to be compared with baselines. For example, DCCAE (Wang et al., 2015) is an autoencoder with CCA that maximizes the correlation between different views. DCCAE also uses different encoders and decoders for each view.
- It needs to be made clear how the splits have been performed. The validation set should be used only for hyperparameter search or checkpoint selection. The final performance must be reported on the hold-out test set.
- It needs to be shown how improvements in reconstruction with regularization improved interpretability. Consider performing analysis with qualitative examples of latents.

Wang, Weiran, et al. "On deep multi-view representation learning." International conference on machine learning. PMLR, 2015.

**Questions:**

- Have you ensured that splits do not have overlapping subjects for EEG data?
- Looking at Figure 5, the model performs as Multi-View models would perform (e.g., SimSiam, SimCLR). The difference is you have a separate Encoder and Decoder, like in DCCAE, instead of multi-view input. Looking at Figure 4, you have a much more diverse representation, or it is because the SAE learned different subspaces. Do you think your regularization will reduce the diversity of the features learned from the data?

---

> ### Author Response · Authors · 2024-11-21
>
> Thank you for your insightful review.
>
> We suggest that DCCAE and MFR are not directly comparable. DCCAE leverages multiple input views, but MFR uses a single view as the input to all autoencoders. For example, in Section 2 of Wang, Weiran, et al. (2015) they refer to “paired observations from two views”. In MFR, we essentially create different views of the input by varying the sparsity constraint on the hidden state. It’s not clear to us how we would apply DCCAE to our problem setting, or MFR to the DCCAE setting, as our understanding is that they assume different problem settings (multiple VS a single input view).
>
> We believe that the baseline used in our paper is the most appropriate baseline, as it is a common architecture for autoencoders trained on language model activations.
>
> We agree that the requested qualitative analysis of latents to gauge the positive effect of improved reconstruction loss on interpretability is an important experiment, and commit to having the following experiment complete for a camera-ready revision:
> 1. Collect data on the activation of SAE features on tokens for the GPT-2 SAEs
> 2. Task humans with explaining the features based on the data from (1)
> 3. Task humans with predicting the activations of SAE features on unseen tokens given the explanations in (2), where more accurate predictions of the activations on unseen tokens indicate more interpretable features
>
> At the reviewers request we could perform this experiment using a language model such as in [Bills et al.](https://openaipublic.blob.core.windows.net/neuron-explainer/paper/index.html), however as mentioned in Section 2.2, recent work has brought this evaluation into question, showing a lack of causal relevance of language model feature descriptions. Ultimately we feel skeptical that there is a good method of evaluation for the interpretability of SAE features.
>
> Per your questions:
> 1. We have ensured there are no overlapping subjects for the EEG data.
> 2. We intend to lose feature diversity across multiple SAEs, as we encourage the learned features to be similar. However, we do not expect to lose feature diversity in a given SAE.

---

> > ### Author Response · Authors · 2024-11-25
> >
> > Thank you for your again for reviewing our submission. As the rebuttal period has nearly concluded, we wanted to kindly follow up to see if we had addressed some of the points in your review, or if we could provide any further clarification.

---

> > > ### Comment · Reviewer_S2HV · 2024-11-25
> > > **Rebuttal Response**
> > >
> > > Thank you for clarifying some of my concerns. I increased the score to "5. marginally below the acceptance threshold".
> > > However, there are two main questions left:
> > > 1) It is unclear how the improved reconstruction improves interpretability, as reconstruction usually saturates where the loss stops improving significantly.
> > > 2) Why is the inductive bias of similar features reasonable? Is there a trade-off for diversity?

---

> ### Author Response · Authors · 2024-11-30
>
> Thank you for revising your score, we are glad we were able to provide some clarification.
>
> We acknowledge that point 1 is important to address. To ensure this is addressed in a camera-ready revision, we would train many probes on the activations of GPT2-Small, each to detect the activation of a particular feature of the input (this would involve estimating a large quantity of features learned by GPT2-Small). We would then train probes to detect the same features in the outputs of SAEs trained with and without MFR. If SAEs trained with MFR learn more features of the input, more of the probes should successfully train on the MFR SAE outputs than the baseline SAE outputs. This experiment would follow Section 4.2 of [Gao et al.](https://arxiv.org/pdf/2406.04093), and was previously used to justify an architectural improvement in SAE training.
>
> For point 2, we believe that Section 3 of our paper establishes why a bias for similar features is reasonable. This is because of the correlation we find between the similarity of features across SAEs, and the similarity of features with the input features. Empirically, this correlation can be leveraged to train SAEs the achieve a lower reconstruction loss, which we believe is further evidence that this is a good inductive bias.
>
> While there may be reduced diversity in the features learned by all of the SAEs (i.e., the same features should appear across each of the SAEs), we do not believe a given SAE should have less diverse features as a result of being trained with MFR (i.e., the same features should not appear multiple times in a given SAE), as we do not see a component of MFR that incentivizes this.
>
> Thank you again for your engagement throughout the review process.

---

### Meta-Review · Area_Chair_Joet · 2024-12-14

**Metareview:**

This paper presents a regularization method, referred to as "mutual feature regularization for improving feature learning by encouraging sparse autoencoders trained in parallel to learn similar features. The main claim here is to enhance neural network interpretability.  The paper is easy to read such that intuitions and related work are well described.  The most critical concern raised by all reviewers is that while enhancing interpretability is the main claim, it is not clear how the improved reconstruction improves the interpretability. In addition, experiments on real-world data is not convincing. When the model is applied to real-world data, the authors did not verify new interpretations, i.e.,  what does the proposed model tell us about EEG data that the baseline did not. Therefore, the paper is not recommended for acceptance in its current form. I hope authors found the review comments informative and can improve their paper by addressing these carefully in future submissions.

**Additional Comments On Reviewer Discussion:**

During the rebuttal period, one reviewer increase his/her overall score a little bit. However, all of reviewers agree that the experiments on real-world data do not provide any new insights over baseline methods.  It is not clear how the improved reconstruction is translated into better interpretability. Since the main issue claimed in this paper is to enhance the NN interpretability, the future submission should clarify this benefit by experiments on real-world data.

---

### Decision · Program_Chairs · 2025-01-22

Reject